# Improving the Aerobic Capacity in Fingerlings of European Sea Bass (*Dicentrarchus labrax*) through Moderate and Sustained Exercise: A Metabolic Approach

**DOI:** 10.3390/ani14020274

**Published:** 2024-01-16

**Authors:** Miquel Perelló-Amorós, Jaume Fernández-Borràs, Shengnan Yu, Albert Sánchez-Moya, Daniel García de la serrana, Joaquín Gutiérrez, Josefina Blasco

**Affiliations:** Department of Cell Biology, Physiology and Immunology, Faculty of Biology, University of Barcelona, Av. Diagonal 643, 08028 Barcelona, Spain; miquelperelloamoros@gmail.com (M.P.-A.); jaume.fernandez@ub.edu (J.F.-B.); yu1107sn@gmail.com (S.Y.); alsanchezmo@ub.edu (A.S.-M.); garciadelaserrana@ub.edu (D.G.d.l.s.); jgutierrez@ub.edu (J.G.)

**Keywords:** sustained swimming, aerobic scope, mitochondrial proteins, nitrogen fractionation, muscles, sea bass

## Abstract

**Simple Summary:**

Based on the results of swimming tests in groups of fingerling sea bass, we selected an appropriate speed of 1.5 BLs^−1^ to apply to sea bass farming; it represents 1/3 of their maximum aerobic speed. Sea bass fingerlings maintained for 6 weeks at this velocity do not impair their growth nor modify the composition of red and white muscle, but they expand their aerobic capacity and show several metabolic changes, especially in white muscle, related to the improvement of their use of nutrients. At the end, fish acquired a more-aerobic phenotype, which leads to greater robustness of the individuals, therefore, increasing their resistance capacity in a wide range of environments without compromising production.

**Abstract:**

Sustained swimming induces beneficial effects on growth and energy metabolism in some fish species. However, the absence of a standardized exercise regimen that guarantees an optimal response to physical activity is due to the anatomical, behavioral, and physiological differences among species, and the different conditions of tests applied, which are especially notable for the early stages of cultured species. The objective of this study was to assess the growth and metabolic responses of European sea bass submitted to continuous and moderate exercise exposure, selecting a practical swimming speed from swimming tests of groups of five fingerlings. The exercise-effects trial was carried out with 600 sea bass fingerlings (3–5 g body weight) distributed in two groups (control: voluntary swimming; exercised: under sustained swimming at 1.5 body lengths·s^−1^). After 6 weeks, growth parameters and proximal composition of both muscles were not altered by sustained swimming, but an increased synthetic capacity (increased RNA/DNA ratio) and more efficient use of proteins (decreased ΔN^15^) were observed in white muscle. The gene expression of mitochondrial proteins in white and red muscle was not affected by exercise, except for *ucp3*, which increased. The increase of UCP3 and Cox4 protein expression, as well as the higher COX/CS ratio of enzyme activity in white muscle, pointed out an enhanced oxidative capacity in this tissue during sustained swimming. In the protein expression of red muscle, only CS increased. All these metabolic adaptations to sustained exercise were also reflected in an enhanced maximum metabolic rate (MMR) with higher aerobic scope (AMS) of exercised fish in comparison to the non-trained fish, during a swimming test. These results demonstrated that moderate sustained swimming applied to sea bass fingerlings can improve the physical fitness of individuals through the enhancement of their aerobic capacities.

## 1. Introduction

Several studies demonstrate that keeping fish to a specific physical activity or swimming exercise leads to a higher growth rate, often coupled with an improved food conversion ratio, FCR [1,2,3]. Moreover, growing fish exposed to water currents exhibit less aggressive behavior and lower levels of stress hormones in the blood, resulting in lower metabolic rates and energy saving [4]. However, these results are not universally conclusive because of the great physiological and behavioral diversity among species. Growth enhancement has been predominantly observed in salmonids such as rainbow trout [5,6,7], salmon [8], and brook trout [9]. Nevertheless, positive effects have also been observed in species from other taxa, such as striped sea bass [10], *Seriola lalandi* [11], or gilthead sea bream [12,13]. Conversely, exercise has either no effect or a negative impact on growth in other species such as cod [14], chinook salmon [15], or European sea bass [16,17]. The variability of those results may be influenced, in part, by differences in the protocols used, individual size, swimming speeds, duration of the studies, and feeding regimens [18]. Furthermore, the intensity of the activity and the species both influence how much energy fuel is used when energy demand varies, as it occurs, during exercise (reviewed by [19]). Furthermore, the type of diet and the availability of nutrients will have a positive or negative impact on growth [20]. In this context, determining the most appropriate swimming speed for each species requires the consideration of multiple factors. Beyond growth improvements, there is a need for tools to assess fish-stock conditions and to improve selection programs with notions such as robustness or resilience [21]. In this regard, exercise can be a good inducer of such characteristics, since, in general, the response to submaximal and chronic swimming is qualitatively like resistance training in mammals, resulting in a more-aerobic phenotype [22,23,24,25]. We have also demonstrated this effect in both juveniles and fingerlings of gilthead sea bream [13,26].

Swim-flume respirometry allows for determining the maximum metabolic rate (MMR) and resting metabolic rate (RMR), two important physiological parameters describing the upper and lower bounds of an organism’s capacity to metabolize energy [27,28,29]. In this type of study, the optimal swimming speed, Uopt, can be determined, which is the speed at which the energy cost is lowest compared to the distance covered. It is the method used by species that migrate long distances [30]. Obviously, maintaining Uopt for extended periods incurs high energy costs because this speed could represent 70–80% of the MMR [31]. Therefore, the identification of speeds where the oxygen consumption is not so high for their application to aquaculture-rearing conditions is necessary. In fact, Carbonara et al. [32] recorded, using an acoustic transmitter, the swimming activity of European sea bass and determined that the mean daily swimming activity corresponds to a third of the critical swimming speed (U_crit_). Another important consideration in the search for a practical speed in aquaculture is to keep in mind that many fish naturally swim in schools. Hydrodynamic benefits of schooling on the critical swimming performance in zebrafish have been reported [33]. Having an approximation of the group oxygen consumption to establish a practical swimming speed in aquaculture for juvenile seabass can be useful.

The European sea bass is one of the most commercially important teleost fish raised in the Mediterranean area. Given the natural environment juvenile sea bass inhabit, proficiency in “sprint”, or “burst type”, locomotion is crucial for this species to hold their position in strong currents and to survive predation of other fish or birds [34]. The aerobic expansibility of sea bass has been the subject of several studies: in response to temperature changes [35], in hypoxic conditions [36], under food deprivation [37], and on individual variability [38]. However, the capability of increasing aerobic scope (AMS) after a period of sustained exercise has not been studied in this species. Beyond this, limited knowledge exists regarding the impact of sustained swimming on seabass fingerlings in terms of growth and metabolism. Adult zebra fish have already been used to show clear evidence of the effects of sustained swimming exposure on their muscle plasticity and swimming performance [39], and we demonstrated the importance of early-life training on the muscle phenotype of gilthead sea bream later in life [26]. Considering the similar response of juveniles and adult fish to swimming activity, applying this earlier in life appears to offer superior benefits for sustainable production. The increase of aerobic capacity may allow for more efficient use of non-protein energy substrates, resulting in a protein-saving effect for growth, as has been observed for rainbow trout [40] and gilthead sea bream [20,26,41]. In gilthead sea bream, this greater energy efficiency responded to changes in the expression of muscle mitochondrial proteins to optimize their function [20], since the mitochondria is the organelle with metabolic plasticity that fits the energy demands [42,43]. As a result, mitochondria constantly adjust to variations in the supply and demands of energy by changing both their morphology (using fusion/fission processes) and their absolute quantity (biogenesis). The purpose of these changes is to accommodate ATP demand (oxidative metabolism) [44], through the different mitochondrial proteins participating in these processes. While numerous studies in mammals have explored the mitochondrial protein response to exercise [44,45,46,47], equivalent studies in fish remain scarce. We investigated this aspect in gilthead seabream. In our earlier research, we found that in fingerlings of this species, the effects of moderate swimming on mitochondrial proteins related to energy metabolism were higher in the white muscle than in the red muscle [20]. Furthermore, because the primary dietary nutrients can be used as energy sources, we found in the same research that their balance altered mitochondrial regulation. However, exercise corrected the nutritional imbalances and improved their redox status [48]. To our knowledge, there are no equivalent studies on sea bass.

Our group has been using the stable isotope (^15^N and ^13^C) analysis technique, CSIA, to determine the use of nutrients and/or reserves in underfeeding and culture conditions [13,20,26,49,50]. We have also been able to verify that exercise in gilthead sea bream reduces nitrogen fractionation in muscles, reflecting higher protein retention and, therefore, enhanced growth in both juveniles [13] and fingerlings [26]. Nitrogen fractionation serves as a robust marker of protein balance [51] and growth rate. Since the aerobic capacity of an organism depends, in part, on the specific demands of each tissue in response to exercise and the energy available from their diet and reserves, we postulate that the isotopic fingerprinting of sea bass exposed to exercise should likewise represent the set of alterations and metabolic changes brought on by exercise.

The main objective of the present work has focused on how exercise can affect the growth and metabolism of the European sea bass, with practical-application purposes for its production. Firstly, the oxygen consumption and the energetic cost of swimming in groups of five fingerlings of sea bass have been quantified in a swimming test to find a practical speed to apply in culture conditions. Secondly, a growth trial using this moderate swimming activity during six weeks has been carried out for fingerlings of sea bass to characterize the growth performance, the recycling of energy reserves using the isotopic levels (δ^13^C and δ^15^N), the expression of mitochondrial proteins, and the activity of two energy-metabolism enzymes in both white and red muscles. Finally, the effect of sustained swimming after 6 weeks of training on the aerobic capacity of individual fish has been determined using swimming tests.

## 2. Material and Methods

### 2.1. Fish Collection and Maintenance

Six-hundred and forty-five sea bass fingerlings (3–4 g b.w) were obtained from a hatchery (Piscimar, Borriana, Spain), and they were reared in the facilities of the Faculty of Biology (University of Barcelona, Barcelona, Spain). Fish were randomly distributed into 8 fiberglass tanks of 200 L, equipped with a semi-closed recirculation system with mechanical and biological filters at 20 °C, and with 12 L:12 D photoperiod regime, and a 35% renewal of seawater each week. The initial stock density was of 1.5 kg·m^−3^ (78 fish per tank). All fish were slightly anaesthetized and measured (body weight and total length). Twenty-five fish of each tank were fitted with a passive integrated transponder (PIT) tag (Trovan Electronic Identification Systems, Madrid, Spain) near the dorsal fin to allow subsequent identification and individual monitoring. Fish were fed with a commercial diet (Inicio plus 868, Biomar Iberia, Palencia, Spain), using automatic feeders and the daily ration of 4% in respect to the total biomass of each tank. The total daily ration was divided into three meals.

### 2.2. Procedure for a Swimming Test

A swim tunnel with a chamber section of 10L (38 × 10 × 10 cm) (Loligo System^®^, Viborg, Denmark) was used to measure the oxygen consumption at different water-current velocities. The swim-tunnel respirometer was thermoregulated at 20° ± 0.5 °C and connected to a 150 L auxiliary tank provided with biological, mechanical, and UV filtration systems. Water flow was generated using a propeller connected to a variable-speed electrical motor. A flush pump allowed the exchange of water between the swim-tunnel respirometer and the auxiliary tank. Water oxygen saturation in the chamber section was measured with an oximeter (Witrox1, Loligo System^®^, Viborg, Denmark) connected to the PC via Bluetooth 2.0. A data-acquisition system recorded every second’s oxygen saturation in the respirometer. The water speed to motor voltage output relationship was established by measuring flow (Flowatch^®^, JDC electronics SA, Yverdon-les-Bains, Switzerland). A laminar flow was induced by passing the water through a honeycomb section placed upstream and a mesh screen located at the downstream end of the chamber.

#### Determination of a Practical Swimming Speed Using a Group-Swimming Test

A group of five fish that were randomly selected, with similar body weights (10.20 ± 0.07 g) and sizes (8.20 ± 0.04 cm), were transferred to the swim-tunnel system for each trial (5 trials) at an initial current speed of 4 cm·s^−1^ (0.5 BL·s^−1^) and rested during an acclimation period (over 30 min) until they observed that oxygen-consumption stabilized. After that, the swimming trial was characterized by 15 min cycle times where the flow velocity was increased progressively in increments of 0.5 BL s^−1^, starting at 0.5 BL s^−1^, until one of the five fish could no longer swim against the flow, showing fatigue. Fish swam at each speed for a 15 min cycle, which consists of 5 min (open circuit) with a continuous oxygen supply and 10 min with the oxygen cut off (closed circuit) to register oxygen-content decay. A recovery period of 30 min at 0.5 BL s^−1^ was carried out after fatigue, and the oxygen consumption was also registered. After that, the fish were anesthetized, weighed, sized, and returned to the tanks. Maximum metabolic rate (MMR) was estimated from the highest oxygen-consumption values (MO_2_) and the maximal swimming speed when fatigue was reached [52], and expressed in U_crit_. Using all the values of oxygen consumption and water velocities, resting metabolic rate (RMR) can be estimated from the minimum oxygen consumption at the minimum swimming speed [53,54]. The decline of oxygen content was expressed using the distance covered and the cost of transport (COT) for each swimming speed:COT = (Δsat(t) × mgO_2_)/m·Δd
where Δsat(t) is the % decline in oxygen saturation during the measurement interval; mgO_2_ is the amount of oxygen in mg per % at saturation under the given conditions, taking into consideration the temperature, the salinity, and the atmospheric pressure of water in every second; m is the body weight of fish in kg; and Δd is the covered distance in meters.

The practical swimming speed was established at 35% of U_crit_ near 1/3 of the maximal velocity [32,55], being the U_crit_:U_crit_ = U_i_ + (t_i_/t_ii_)·U_ii_
with U_i_ being the highest velocity maintained for the entire prescribed period; U_ii_, the velocity increment; t_i_, time to fatigue at final velocity level; and t_ii_, time for which each velocity level is maintained.

### 2.3. Growth Trial under Sustained and Moderate Swimming Speed: Experimental Set-Up and Protocol

Fish were randomly distributed in eight 200 L fiberglass tanks. In four of them, a circular and uniformly distributed water flow of 700 L/h induced by the water coming out of small holes on a pipe connected to the water entrance forced the fish to achieve sustained swimming (Exercised, Ex). A cylindrical tube in the center of these tanks prevented the fish from entering the area of lower velocity. The swimming velocity was adjusted to 1.5 body lengths per second (BL·s^−1^), being the same in the water column, as was determined with a low-speed mechanical flow meter (General Oceanics Inc., Miami, FL, USA). The flow was also measured at three points from the periphery to the center of the tank, observing a 30% reduction at the maximum. With the exception of being maintained in typical raising conditions and having a vertical water inlet and a water flow of 350 L/h (Control, Ct), the other four tanks were identical. Sea-water composition was the same and the renovation guaranteed optimal levels of physical parameters. During the six weeks of the experimental period, all groups were fed three times a day at 4% ration. Body weight, standard length, and total length were measured at 22 and 44 days after the start of the trial. The specific growth rate was calculated for each PIT-tagged fish [SGR = 100 × (ln final body weight − ln initial body weight/time)].

At the end of the experimental period, a swim-to-fatigue test was tested individually on 6 trained and 6 untrained fish, following the same experimental procedure as above. As before, the aerobic scope (AMS) was calculated from the difference between resting metabolic rate (RMR) and maximum metabolic rate (MMR), and COT was also calculated.

### 2.4. Preparation of Tissue Samples

Twelve fish from each treatment (three randomly selected fish per tank, four tanks per experimental condition) were weighed, measured, and put under anesthesia at the conclusion of the experiment. The spinal cords of the animals were severed, and they were then eviscerated. The weights of the liver and mesenteric fat were determined in order to calculate the hepatosomatic index (HSI) and mesenteric fat index (MFI). Red and white skeletal muscle samples were obtained, and they were kept in liquid nitrogen until additional examination. The experiments complied with the guidelines of the Council of the European Union (2010/63//EU), the Spanish government (RD 53/2013), and the University of Barcelona (Spain) for the use of laboratory animals (DAAM 9436).

### 2.5. White Muscle Proximate Composition and Isotopic Composition Analysis (δ^15^N and δ^13^C)

White muscle samples were homogenized in liquid N_2_ using a pestle and mortar to obtain a fine powder for the following analysis. For isotope studies and to evaluate the lipid, protein, glycogen, and water content, aliquots of every sample were obtained. After the samples were dried for 24 h at 95 °C, the water content was measured gravimetrically. Following Folch et al.’s instructions [56], lipid extracts were dried under N_2_, and the total amount of lipids was calculated gravimetrically. Proteins from tissue samples that had been defatted were extracted using 10% (*v*/*v*) trifluoroacetic acid precipitation. The extracts were dried using a vacuum system (Speed Vac Plus AR, Savant Speed Vac Systems, South San Francisco, CA, USA). Assuming 1 g of N for every 6.25 g of protein, the protein content was estimated from the total N content obtained using elemental analysis (Elemental Analyser Flash 1112, ThermoFinnigan, Bremen, Germany).

After alkaline hydrolysis, tissues were boiled with 30% KOH to extract and purify the glycogen [57]. The anthrone-based colorimetric approach was then used to determine the glycogen content [58]. Muscle nucleic acid levels (RNA and DNA) were assessed using UV-based procedures [59]. Nucleotide concentrations were determined by hydrolyzing muscle sample RNA and DNA to nucleotides and measuring the absorbance at 260 nm. The units used to represent nucleic acid concentrations were μg of RNA or DNA per mg of wet tissue. The protein concentration was also measured using an aliquot of supernatant [60].

Tiny tin capsules containing aliquots of the food, white muscle, and their pure tissue components (glycogen, lipid, and protein) were weighed, which varied from 0.3 to 0.6 mg. Samples of feed and white muscle were lyophilized and grounded into a homogenous powder for isotopic analysis; aliquots ranging from 0.3 to 0.6 mg were weighed in small tin capsules. Using a Mat Delta C Isotope Ratio mass spectrometer (Finnigan MAT, Bremen, Germany) connected to a Flash 1112 Elemental Analyzer, samples were examined to ascertain the composition of carbon and nitrogen isotopes. The isotope ratios (^15^N/14N, ^13^C/12C) obtained using isotope ratio mass spectrometry are shown in delta (δ) units, which are parts per thousand, ‰, as follows:δ = [(Rsa/Rst) − 1] × 1000
where Rsa is the ^15^N/^14^N or ^13^C/^12^C ratio of samples, and Rst is the ^15^N/^14^N or ^13^C/^12^C ratio of the international standards (Vienna Pee Dee Belemnite for C and air for N). Measurements were made with ±0.2‰ accuracy using the same reference material that was examined throughout the experiment. The difference between the δ value in the tissue and its equivalent δ value in the diet is used to compute nitrogen isotopic fractionation (Δδ^15^N).

### 2.6. Gene Expression (RNA Extraction and cDNA Synthesis)

Tissue samples (100 mg of white or red muscle) were homogenized in 1 mL TRI Reagent^®^ with Precellys^®^ Evolution Homogenizer and cooled with Cryolys^®^ (Bertin-Techmologies, Montigny-le-Bretonneux, France) at 4–8 °C. After homogenization, RNA extraction was performed following the TRI Reagent^®^ manufacturer protocol. Afterward, a NanoDrop 2000 (Thermo Scientific, Alcobendas, Spain) was used to determine total RNA concentration and purity. Confirmation of RNA integrity was performed in a 1% (*m*/*v*) agarose gel stained with SYBR-Safe DNA Gel Stain (Life Technologies, Alcobendas, Spain). cDNA synthesis was performed like in Salmeron et al. [61], but samples were first treated with DNase I Amplification Grade (ThermoScientific, Alcobendas, Spain), and, following that, retrotranscription was performed with a Transcriptor First Strand cDNA Synthesis Kit (Roche, Sant Cugat, Spain) according to the manufacturer’s recommendation.

As a technique to analyze gene expression, we performed a quantitative polymerase chain reaction from the cDNA samples using a previous cDNA-synthesis step (RT-qPCR). The q-PCR was performed with iTAQ Universal SYBR^®^ Green Supermix (Bio-Rad, El Prat de Llobregat, Spain), in Hard-Shell^®^ 384-well PCR plates (Bio-Rad, El Prat de Llobregat, Spain), and CFX384TM Real-Time System (Bio-Rad, El Prat de Llobregat, Spain). The q-PCR program was of 3 min at 95 °C; 39× (30 s at 95 °C, 30 s at primer melting temperature and fluorescence detection); 5 s at 55 °C; final fluorescence detection; and temperature rising to 95 °C until 4 °C maintenance. After the program ended, wrong wells were eliminated using CFX Manager^TM^ 3.1 software (Hercules, CA, USA) before calculating the relative gene expression values with the same software.

The sequences, melting temperatures, and accession numbers of the primers used in the real-time quantitative PCR analysis are displayed in Table 1. The following genes were used as reference genes: EF1a (Elongation factor 1-alpha 1), RPL13 (60S ribosomal protein L13), RPL17 (Ribosomal protein L17), and FAU (40S ribosomal protein S30). They were analyzed, and the combination of the most stable ones was used to calculate the relative expression of the genes of interest following the Pfaffl method [62]. The stability of the reference genes (assessed with the geNorm algorithm) and the relative expression to the geometric mean of the reference genes were calculated with the CFX Manager^TM^ 3.1 software.

### 2.7. Enzymatic Activities

Samples of white and red muscle (250 mg or 100 mg) were powdered and dissolved with 2 or 1 mL of buffer solution, respectively, with the stabilizing solution:detergent (1:1). The samples were homogenized with Precellys^®^ Evolution Homogenizer, cooled with Cryolys^®^ at 4–8 °C, centrifuged (15 min at 1000× *g*), and aliquots of the supernatant were distributed in Eppendorfs and stored at −80 °C for the subsequent enzymatic analyses.

CS activity analyses were performed with the Citrate synthase Cayman N 701040 kit following the method described by Srere [63]. The absorbance increases at 412 nm of DTNB reagent, using oxalacetic acid as the substrate, was measured. The CYTOC-OX1 kit (Sigma-Aldrich, Tres cantos, Madrid, Spain) was used to measure COX activity. This colorimetric test quantifies the reduction in ferricytochrome c absorbance that results from COX oxidizing the latter. Enzyme activities were measured in milliunits (mUI·g^−1^), which represent one unit of the quantity of enzyme required to convert one μmol of substrate per minute. Bradford’s technique was utilized to assay the protein concentration using additional aliquots of the supernatant.

### 2.8. Protein Expression Using Western Blot

The processing of the samples for protein extraction, quantification, and Western blot analysis of a series of energy metabolism and mitochondrial proteins was carried out as previously established in gilthead seabream [20]. Briefly, using the Precellys^®^ Evolution connected to a Cryolys chilling system (Bertin Technologies, Montigny-le-Bretonneux, France), protein was extracted from 100 mg of skeletal white muscle in 1 mL of RIPA buffer supplemented with phosphatase (PMSF and NA3VO4) and protease inhibitors (P8340, Sigma-Aldrich). Bradford’s technique was used to determine the soluble-protein concentration. Twenty micrograms of the soluble protein fraction were diluted in a loading buffer with SDS and -mercaptoethanol, heated to 95 degrees Celsius for five minutes, and then separated on a polyacrylamide gel (sixteen percent for COX4, fifteen percent for CS and UCP3). After electrophoresis, the proteins were immersed in methanol for an overnight period before being transferred to Immobilon^®^ PVDF-FL 0.2 µm transfer membranes (Merck Millipore Ltd., Tullagreen, Cork, Ireland). The total transferred protein was determined with 5 min incubation with REVERT^TM^ Total Protein Stain (LI-COR, Lincoln, NB, USA). The signal was read at 700 nm using the Odyssey Fc Imaging System (LI-COR). After total protein quantification, the membranes were blocked with Odyssey Blocking Buffer (diluted 1:1 in TBS) (LI-COR) for 1 h at room temperature before being incubated overnight at 4 °C and agitated with the corresponding primary antibody diluted in blocking buffer + 0.05% Tween20. The primary antibodies used were purchased from ABCAM (Cambridge, UK) as follows: rabbit polyclonal anti-Cox IV antibody (ab16056) 1/1000, rabbit polyclonal anti-citrate synthase antibody (ab96600) 1/2000, rabbit polyclonal anti-UCP3 antibody (ab180643) 1/500 for white muscle (WM), and 1/1000 for red muscle (RM). The molecular weight of the bands verified the antibodies’ cross-reactivity with sea bass. Following TBST washing, the membranes were incubated with matching secondary antibodies diluted at a 1:10,000 dilution in blocking buffer + 0.05% Tween20: IRDye^®^ 800CW Goat anti-Rabbit (925-32211), Li-Cor, Lincoln, NB, USA). After incubation, the membranes were washed with TBS-T and the fluorescence of the immunoreactive bands was measured at 800 nm using the Odyssey Fc Imaging System (LI-COR). The Appendix A compiles the raw images taken for the Western blot analysis.

### 2.9. Statistical Analysis

The results of tissue muscle composition, nucleic acid contents, enzyme activities, gene and protein expressions, and the isotopic analyses are presented as mean ± standard error of the mean (SEM). Initially, Shapiro–Wilk test followed by Levene’s test were used to check the normal distribution and homogeneity of variances, respectively. Secondly, a nested ANOVA was run to discard tank effects in any of the variables analyzed. As no tank effects were found, each group of 3 fish per 4 tanks resulted in an *n* of 12 individuals per condition [Control (Ct) vs. Exercise (Ex)]. Then, Student’s *t*-test was used to compare the values of all variables between CT and EX groups. A significant threshold of *p* < 0.05 was established. The SPSS software, version 22.0 (IBM SPSS Statistics for Windows, Version 22.0, IBM Corp., Armonk, NY, USA), was used for all of the analyses.

## 3. Results

### 3.1. Group-Swimming Test for Sea Bass Fingerlings

The swimming performance and oxygen consumption of sea bass fingerlings were determined at the rearing conditions of the control fish in groups of five individuals (mean body weight of 10.20 ± 0.07 g, and body length 8.20 ± 0.04 cm). As is shown in Figure 1A, a linear relationship of the MO_2_ of the entire group with swimming speed was found with increasing water velocity between 0.5 and 4 BL·s^−1^. At 4.5 BL·s^−1^, fatigue was observed in some fish in each group, which marked the end of the test and was considered the U_crit_ in this condition. The RMR extrapolated from the linear MO_2_–speed relationship considered at zero swimming speed was 434 mg O_2_·kg^−1^.h^−1^. At 4.5 BL·s^−1^, MO_2_ was increased by 111%, whereas at 1.5 BL·s^−1^, it was 37% of RMR.

Likewise, COT (expressed by mg O_2_·kg^−1^.km^−1^) was calculated (Figure 1B). The vertical line at the speed 1.5 is the nearest point of the curve from the origin (X = 0; Y = 0), representing a change of the slope of the water with a speed lower than 1.5 and those faster than 1.5 BL·s^−1^. This was the swimming speed proposed for the next growth trial, a moderate and aerobic swimming speed that represents nearly 1/3 of the MMR.

### 3.2. Growth Trial at Moderate Swimming Speed

#### 3.2.1. Growth Performance, Proximal and Isotopic Composition

The effects of a sustained moderate swimming regime at 1.5 BL·s^−1^ for 6 weeks on body-growth parameters and somatic indices in sea bass fingerlings are shown in Table 2. Both groups of fish were maintained on a fixed food ration of 4%. Under these conditions, no significant differences were observed in the final weight; although the average value was 8% lower for exercised fish than for the control fish, individual variability makes the difference insignificant. Likewise, there are no differences in the condition factor (CF) or the specific growth rate (SGR). Similarly, the hepatosomatic (HSI), muscle somatic (MSI), and mesenteric fat (MFI) indexes did not differ significantly between both groups; although, in the case of exercised fish, the MSI average value was 4.8% higher.

Regarding the proximal composition of the WM (Table 3), no differences were observed in protein, glycogen, or lipid content between both groups; although, the low content of lipids normally found in white muscle was slightly increased in exercised fish. For the nucleic acids, RNA content increased and DNA content decreased in the exercised fish, and the combined effect of both changes determined a significant increase in the RNA/DNA ratio in the exercised fish (*p* < 0.05).

The results of the isotopic analysis in WM are shown in Table 4. The δ^15^N value was significantly lower in the exercised fish (*p* < 0.005). This caused the fractionation of ^15^N in white muscle, and the difference between the δ^15^N tissue and the δ^15^N diet of the exercised group was also lower (*p* < 0.005).

#### 3.2.2. Mitochondrial Proteins in White and Red Muscle: Gene and Protein Expression and Enzyme Activities

Figure 2 shows the activities of the COX and CS enzymes of WM. No significant differences were observed in COX or CS activity between both groups of fish, but the CS activity of the exercised fish was lower, which determined that the ratio of COX/CS activity increased significantly in the exercised fish (*p* < 0.05).

The gene expression values of mitochondrial proteins (*cox4a*, *cs*, *pgc-1α*, *ucp3*, *fis1*, and *mit1*) in both muscles are shown in Figure 3. The exercise significantly increased *ucp3* gene expression in both muscles. *pgc-1α* and *fis1* increased in WM (Figure 3) and *cox4a* and *cs* decreased in RM (Figure 3), but these changes were not significant. The changes in CS, COX4, and UCP3 protein expression in both muscles are shown in Figure 4. A significant increase (*p* < 0.01) of COX4 and UCP3 protein expression was observed in the WM of exercised fish (Figure 4). In the RM of exercised fish (Figure 4), only the protein expression of CS increased, but not significantly (*p* = 0.058). For this reason, the COX4a/CS ratio of exercised fish significantly increased by 1.75 times in their WM and decreased by 0.5 times in their RM.

Oxidative-stress-related genes’s expressions (*sod*, *cat*, *gpx*, *gshr*) for WM and RM are shown in Figure 5. In WM, the exercise induced higher gene expression in catalase, while no change was observed in the expression of these genes in RM.

### 3.3. Effect of Moderate Swimming Training on the Aerobic Capacity of European Sea Bass

After the exercise trial, the oxygen consumption of the fish in training (exercised) or non-training (control) at different swimming speeds was measured (Figure 6A), and the values were adjusted to a polynomic curve. Non-differences were observed in RMR between the two groups. However, MMR in the exercised group was significantly higher (*p* < 0.01) than in the control group (Table 5). This agrees with AMS which, although not significant, showed a tendency to be higher in the exercise group. When COT for both groups is represented (Figure 6B), the values are adjusted to a potential decay curve, but no differences were observed between the two groups.

## 4. Discussion

### 4.1. Finding a Practical Swimming Speed for Sea Bass Farming

Some fish submitted to moderate swimming activity improved their growth rates [12,64,65], but other studies report no effect or even negative impacts on growth performance [66,67]. Since aerobic swimming has a metabolic cost, which varies depending on swimming speed, the question to be resolved is which speed is the most appropriate to stimulate or promote growth [18]. In some species, such as salmonids, improvements in growth have been observed when fish were subjected to the optimal swimming speed, where the lowest transport cost is achieved [1,68], but this is not the case for sea bass. Juvenile sea bass males maintained at the optimal swimming speed (69% U_crit_) for 10 weeks did not modify their growth, but delayed their testicular development reducing the incidence of sexually precocious males [16]. Despite that the fish are considered to swim at the optimal speed in routine movements [69], there are many discrepancies between the measurement of Uopt and RMR in species that do not have constant and linear swimming, such as cod [70,71] or gilthead sea bream [55]. Then, the use of Uopt during swimming by wild fish is often assumed, but this has not been satisfactorily demonstrated, essentially due to technical limitations [72]. Furthermore, although the optimal speed may represent the lowest transport cost, the fact that it is 70–80% of MMR [73] makes it suitable for long migrations but not for maintaining it for prolonged periods because it implies high energy costs. Therefore, looking for speeds where oxygen consumption is not so high, but closer to the daily metabolic rate, leads us to determine practical application speeds for aquaculture. Moreover, an important consideration in the search for a practical speed for aquaculture is to keep in mind that the studies searching for a U_crit_ of fish have typically been measured on individuals given a step-based test that ends in fatigue. However, in zebrafish schooling improves critical swimming performance, as there are hydrodynamic benefits to schooling [33]. This is the rationale behind measuring the oxygen consumption in groups of five fish to establish a practical velocity for juvenile seabass. The results of our grouped-fish swimming test showed that a speed of 1.5 BL s^−1^ represented 35% of U_crit_ (at 4.5 BL·s^−1^). These results are in agreement with those observed in this species using an acoustic transmitter to record swimming activity [32], where the mean swimming activity levels corresponded to speed being one third of the U_crit_. In gilthead seabream, the optimal oxygen consumption was also obtained at 1.5 BL s^−1^, corresponding to 30–40% of their U_crit_, where aerobic metabolism is ensured [55]. All these results demonstrate that fish can swim at lower speeds that they can sustain for days, weeks, or even months. Thus, the chosen speed, as a practical velocity for fingerlings of sea bass, was 1.5 BL s^−1^, where the optimal MO_2_ of swimming was estimated at about 1/3 of MMR.

### 4.2. Growth Trial at Moderate Swimming Speed

#### 4.2.1. Growth Performance, Proximal and Isotopic Composition

Growth improvement through exercise has been observed for different species, with a close correlation to the specific physical-activity profile of each species. In this sense, the improvement has been observed especially in pelagic species [2], and, in the case of gilthead sea bream, using the optimal swimming speed [74]. However, in the current study involving sea bass under a regime of moderate and sustained swimming (1.5 BL s^−1^) for 60 days, no significant improvements in growth rate or fillet composition were observed. Despite the lack of significant differences due to individual variability, there was a noteworthy 21% increase in lipid reserves. Higher levels of these preferred substrates in the tissues might allow the fish to achieve higher rates of aerobic work [75]. In juvenile males of seabass maintained at Uopt (using 69% U_crit_) for 10 weeks, the growth was not modified either [16]. No improvement was observed in growth or fillet composition, but rather in muscle texture in sea bass of 300 g body weight that were maintained for 32 days at 1 BL s^−1^ (25% of the U_crit_) [17]. The characteristics of the swimming patterns of sea bass have been the subject of various works, most of them carried out by individual swimming tests in a swim tunnel [76,77]. Luna-Acosta et al. [78] did not find differences between wild and cultivated sea bass (body weight: 270 g) when comparing SMR and MMR, which pointed out that sea bass have strong physiological plasticity, and the domestication and selection process did not affect their swimming ability. Interestingly, sea bass categorized as “slow sprinters” grow faster than those reaching a speed of 11 BL s^−1^ [79]. Although the growth rate did not increase in the present study, indicators of improvement were found such as a greater synthetic capacity of muscles (with greater RNA/DNA ratio) and better protein retention (with reduction in ^15^N fractionation). Both improvements were also found in exercised gilthead sea bream [13,26]. In this sense, the significant reduction of nitrogen fractionation (Δ^15^N) in WM of exercised sea bass would indicate the improvement of the protein balance, as this parameter is known as a good marker of protein turnover and protein-retention efficiency [49,51]. Since the feeding ration was the same in both exercised and control groups, the lower Δ^15^N was due to a better use of nutrients under exercise. Exercise is an energetically demanding condition, but the improvement in the use of diet in exercised fish would compensate for the greater demand, and it does not translate into less growth. Possibly adjusting the diet composition to meet this greater energy demand could be beneficial and result in higher growth as we have observed in gilthead sea bream [20].

#### 4.2.2. Mitochondrial Proteins in White and Red Muscle: Gene and Protein Expression and Enzyme Activities

The regulation of the mitochondrial life cycle in skeletal muscle, from the biogenesis of new mitochondria to the elimination of dysfunctional ones, determines the quantity, quality, and function of mitochondria, which are determinants of metabolism and physical performance [80]. This dynamic process is evident in the mitochondria of skeletal muscle after physical exercise in mammals and fish. A previous study showed alterations in the expression of genes and proteins linked to energy metabolism, mitochondrial fusion, and fission processes in sea bream exercised at 2.5 BL s^−1^ for six weeks [20]. However, the gene expression of most mitochondrial proteins in both WM and RM of fingerlings of sea bass submitted at 1.5 BL·s^−1^ remained unaltered, except for an increased expression of *ucp3* in both muscles. This finding aligns with the results previously observed in gilthead sea bream [20]. The increase of *ucp3* gene expression uncouples oxidative phosphorylation allowing the entry of lipids for the production of β-oxidation intermediates [81]. In this case, a greater production of peroxides, resulting from this increase in the oxidation of substrates in the muscles of exercised sea bass, would be expected, which would induce the observed increase of *cat* gene expression. The hydrogen peroxide produced by SOD is converted into water through CAT, which is mainly located in peroxisomes and in the mitochondria [82]. This increase in *cat* gene expression of WM coincides with an increase in the protein expression of UCP3 in the target muscle that would prevent the overexpression of ROS [81] because of the boosted oxidative metabolism. In addition, the enhancement of the oxidative processes is reflected in the increased COX protein expression in the muscle. While a decrease in COX enzymatic activity and an increase in CS was reported in exercised gilthead sea bream juveniles [26], in the case of sea bass we did not observe changes in the two enzyme activities in either muscle of the exercised fish. Interestingly, our results for sea bass suggest that more COX protein expression does not imply greater activity of this protein in WM. We hypothesize that the increase in UCP3 limits the activity of COX even if there is more COX protein expressed. Despite similar enzyme activities, the ratio of COX to CS was significantly higher in exercised sea bass. This suggests that alterations in mitochondrial size and shape are in response to energy-demand conditions, which is consistent with the findings of other studies [83,84]. The differences in the expression response and activity of metabolic enzymes such as COX and CS between gilthead sea bream and sea bass highlight the remarkable metabolic plasticity of fish, enabling them to adapt to tissue-specific demands in response to varying exercise and energy availability.

In contrast to what was observed in WM, increased expression of the *ucp3* gene in RM was not translated into an increase in UCP3 protein expression, and neither was it followed by an increase in COX protein expression. This suggests that the possible greater oxidation of substrates does not induce an oxidative overload in this tissue, coinciding with what we observed in gilthead sea bream, where an increase in UCP3 protein expression was not observed in RM, despite the increase in lipid oxidation [20] which is the main energy source in this tissue. The lower impact of the oxidation of energy substrates was reflected in the non-response of antioxidant mechanisms in this tissue, as we had seen in the RM of exercised gilthead sea bream, suggesting a high basal buffering capacity [48]. In rainbow trout, AMS and swimming performance are positively related to maximum cardiac-pumping capacity [85]. However, the supply of oxygen to the red muscle is not a limiting factor at the time of exhaustion in trout [86]. Therefore, the oxidative capacity of RM is favored by exercise without problems in its redox status, perhaps due to changes in energy fluxes and metabolism support rather than in properties of muscle function per se, as Claireaux et al. [76] suggested. Consequently, moderate swimming affects mitochondrial proteins linked to energy metabolism more in WM than in RM, which is consistent with findings in gilthead sea bream [20]. Furthermore, sustained moderate exercise causes the white muscle (WM) of sea bass to shift towards a more-aerobic phenotype, which is consistent with the findings of gilthead sea bream [13,26] and other species [22,23,87].

### 4.3. Effect of Moderate Swimming Training on the Aerobic Capacity of European Sea Bass

In order to demonstrate that maintaining moderate and continuous exercise over time influences the aerobic capacity of sea bass, a swimming test was performed on individuals from the two groups of sea bass (exercised and controls). The effects of exercise on MO_2_, and the U_crit_ performance, were very similar to those described previously for this species at this temperature [76]. A tendency toward an asymptote in MO_2_ at the highest speeds was observed, reflecting the initiation of anaerobic swimming. In our study, the absolute U_crit_ values ranged 60 a 72 cm·sg^−1^ and were consistent with those reported in the literature for sea bass [76,77]. On the other hand, the values obtained for relative U_crit_ are higher (3 to 4.2 BL·s^−1^) than those obtained by Alfonso et al. [88], because the weight of fish was much lower in respect to that used by these authors, and an inverse relationship exists between the relative U_crit_ and body weight of sea bass [88,89]. Moderate and sustained swimming training did not cause changes in RMR compared to untrained ones, but it clearly increased the MMR of trained seabass resulting in a tendency to increase their aerobic expansiveness. AMS has been proposed as a measure of whole-organism fitness in various environmental conditions [90,91]. Therefore, obtaining physically fit individuals could improve growth capabilities at an early juvenile stage, as the selective breeding for rapid growth, delayed maturity, and high fecundity appears to have reduced athletic robustness in aquaculture selection procedures [92].

## 5. Conclusions

In conclusion, the results of the present study indicate that maintaining sea bass fingerlings at about 1/3 of the maximum metabolic rate (1.5 BL s^−1^), induces an increase in the aerobic capacity of their muscle, an improvement in their protein balance, and no changes in their growth performance and proximal composition. Therefore, this sustained swimming speed determines an improvement in the physical fitness of juvenile sea bass. This would contribute to rearing robust animals with good exercise performance for use in restocking programs, promoting their survival, and increasing their resistance capacity in a wide range of environments without compromising production.

## Figures and Tables

**Figure 1 animals-14-00274-f001:**
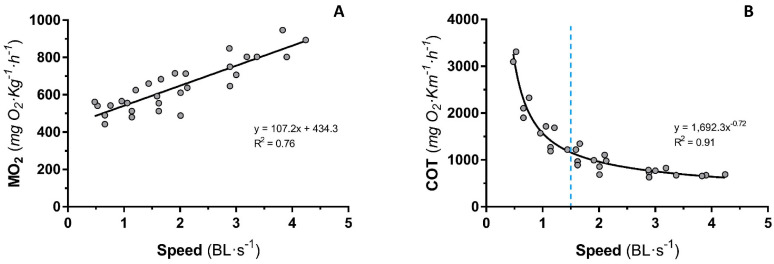
Oxygen consumption (**A**) and cost of transport (**B**) of sea bass fingerlings at different swimming speeds. Each value corresponds to five fish grouped. *n* = 5 test. The vertical line at the speed 1.5 BL·s^−1^ is the nearest point of the curve from the origin (X = 0; Y = 0), representing a change of the slope of the water speed lower than 1.5 and those faster than 1.5 BL·s^−1^.

**Figure 2 animals-14-00274-f002:**
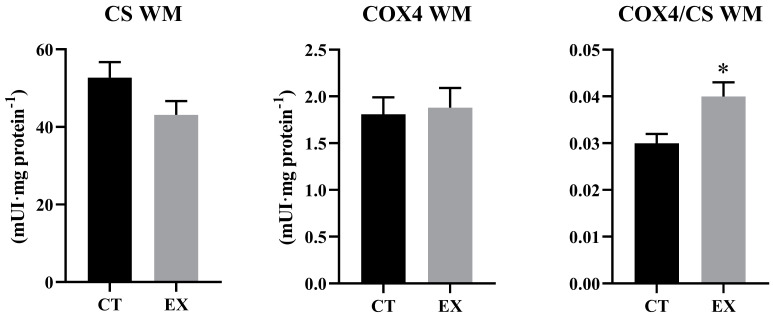
Citrate synthase (CS) and citocrom-c-oxidase (COX) activities in white muscle of sea bass voluntarily swimming (CT) or subjected to sustained swimming (EX). The values represent the mean ± standard error of the mean (*n* = 12). Significant differences according to the Student’s *t*-test: (*) *p* < 0.05.

**Figure 3 animals-14-00274-f003:**
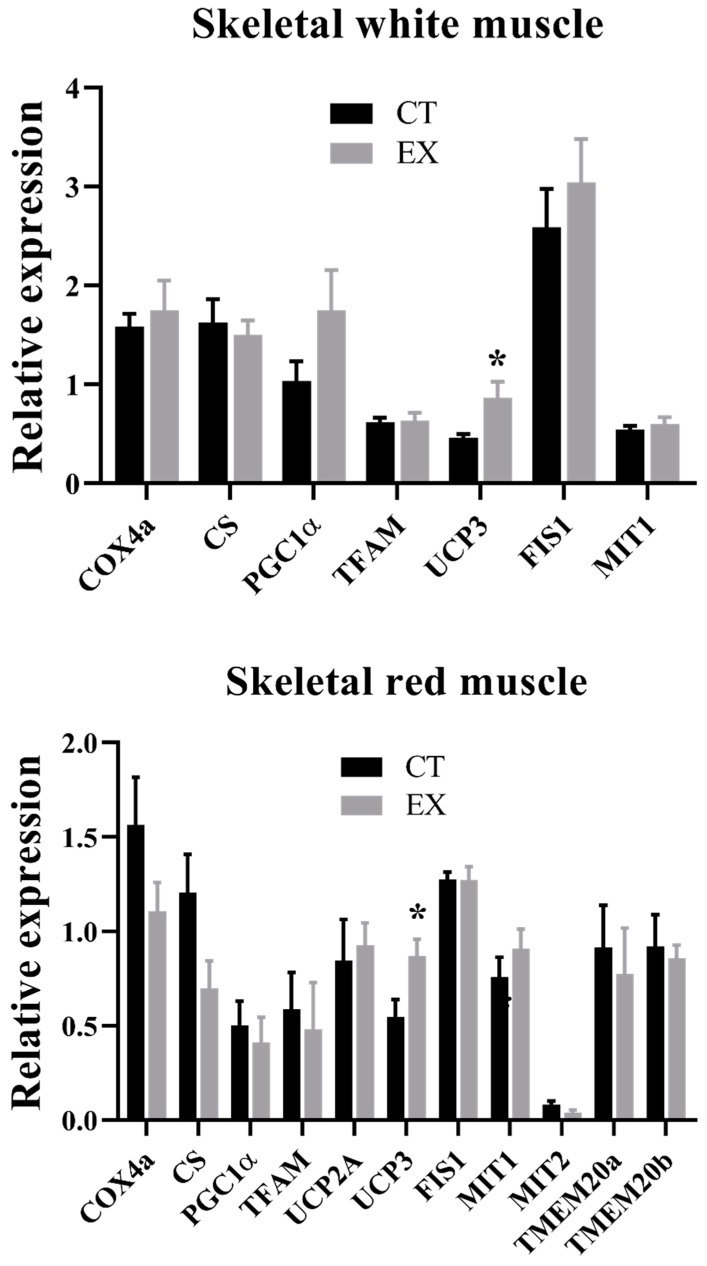
Gene expression of mitochondrial proteins in white and red muscle of sea bass fingerlings voluntarily swimming (CT) or subjected to sustained swimming (EX). The values represent the mean ± standard error of the mean (*n* = 12). Significant differences according to the Student’s *t*-test: (*) *p* < 0.05.

**Figure 4 animals-14-00274-f004:**
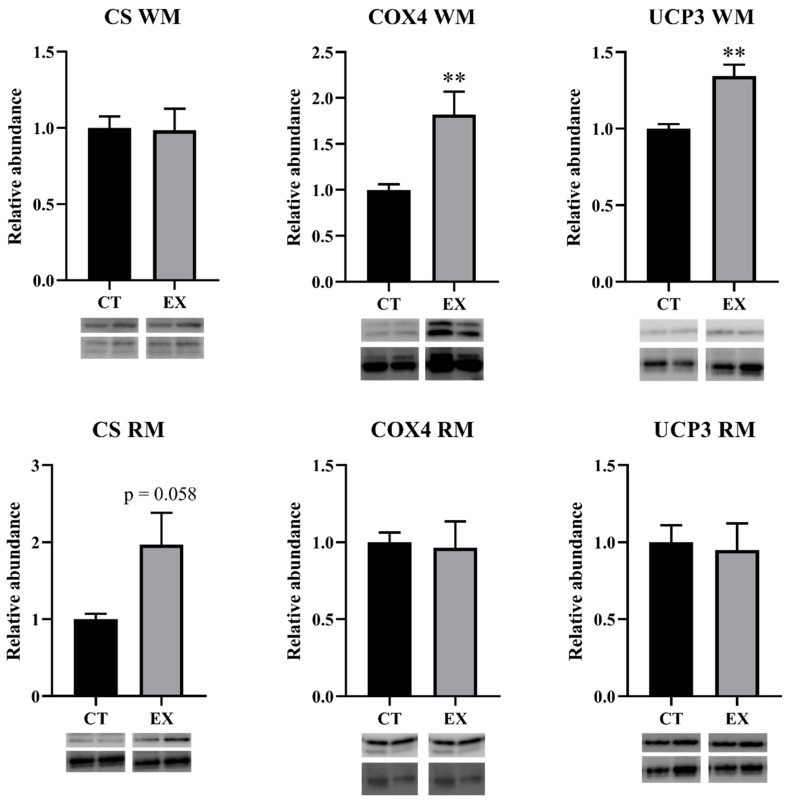
Protein expression of mitochondrial proteins in white and red muscle of sea bass fingerlings voluntarily swimming (CT) or subjected to sustained swimming (EX). Bands were normalized to their RevertTM total protein staining (the corresponding well is shown). The blots for COX4a and CS were carried out using the same membranes that had been split prior to the primary antibody incubation. Student’s *t*-test (unpaired) was used to evaluate pairwise comparisons: ** *p* < 0.01. CS: citrate synthase; COX: cytochrome-c-oxidase; UCP: uncoupling chain protein.

**Figure 5 animals-14-00274-f005:**
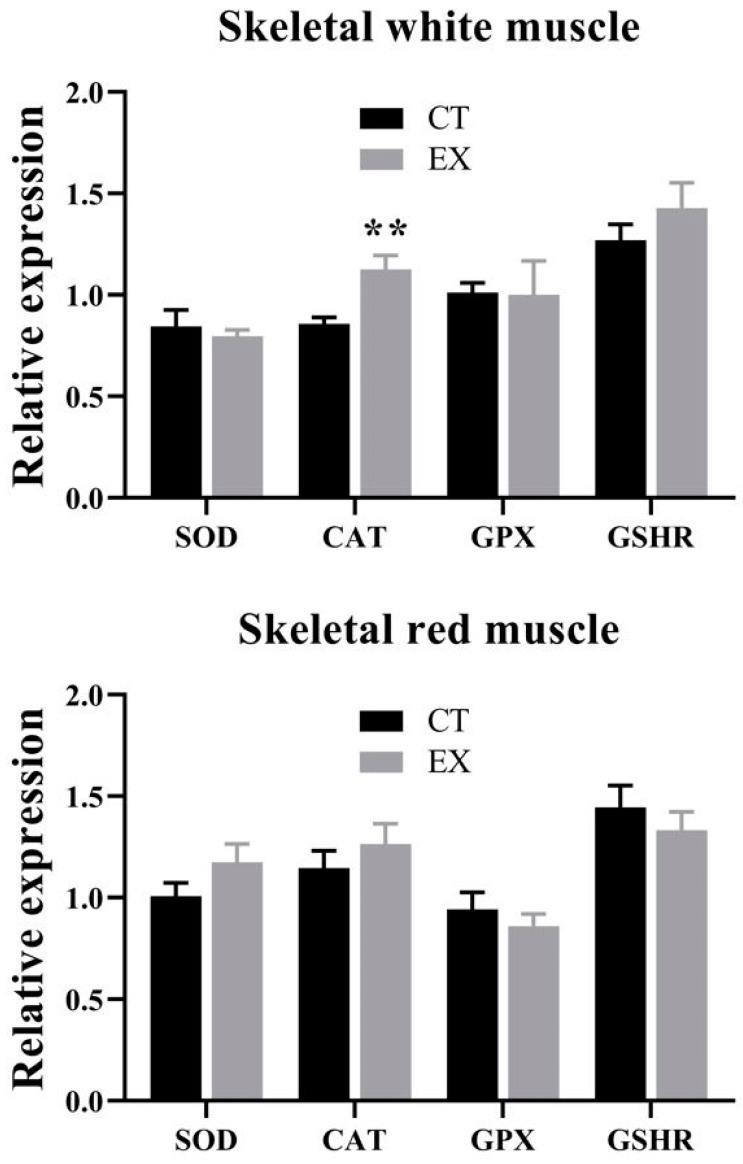
Relative expression of genes associated with oxidative stress in white and red muscle of sea bass voluntarily swimming (CT) or subjected to sustained swimming (EX). Student’s *t*-test (unpaired) was used to evaluate pairwise comparisons. ** *p* < 0.01.

**Figure 6 animals-14-00274-f006:**
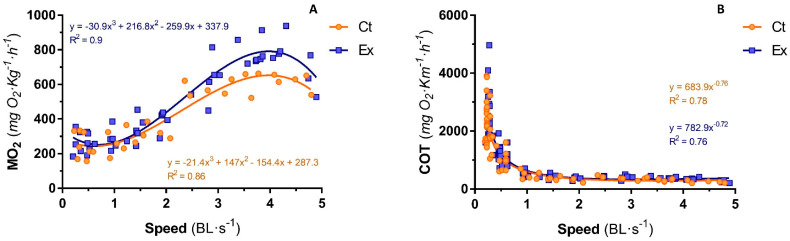
Oxygen consumption (**A**) and cost of transport (**B**) at different swimming speeds in control (orange) and exercised (blue) sea bass. Each value corresponds to individual fish (*n* = 6 for control fish; *n* = 6 for exercised fish).

**Table 1 animals-14-00274-t001:** Primers used for real-time quantitative PCR.

Gene	Sequences 5′-3′	Ta (°C)	Accession Number
*cox4a*	F: CATTGTACCGCATCAGCTTC R: GGCCAGTGAAGCCAATTAAG	60	XM_051380547
*cs*	F: ATTGGGCACAAGGTCTGAAC R: AAGCAGGAAAGCTTGAGACG	60	XM_051425289
*pgc1a*	F: GCAAACTCCCAGCTCAGCTA R: GCTTCACTGGCTTTGGTGTG	60	XM_051429129
*tfam*	F: CCTGCACAAGTACTCTGGCA R: ACGGCTCTTTCTGTTCTGGG	60	XM_051417177
*ucp2a*	F: AAGGAGGAAGGCATTCGTGG R: CCCTGCACCAAAGGCTGATA	60	XM_051401934
*ucp3*	F: TGATGACGGACAACATGCCA R: GGAGCCCAGACGAAGATACG	60	XM_051397440
*mit1*	F: AAAGAGGTGCTCAACTCCCG R: GTTGATGGCGTCCATGATGC	60	XM_051375872
*mit2*	F: TCAGGAAGCTCCATGTGCTG R: CACTGACGAGGAACCAGCTT	60	XM_051426021
*fis1*	F: TTGTTGAAGGGAGCCGTCTC R: AGTCTGTAGTTGGCCACTGC	60	XM_051429174
*Tmem20a*	F: ACCACCTGACCAATGCGATT R: GTGGGCAGTTTTGTGAGCAG	60	XM_051429207
*Tmem20b*	F: TGCTGGCTCAGGGAGACTAT R: TGGTGAGCAGCATCTGGAAG	60	XM_051381355
*sod*	F: GTTGGAGACCTGGGAGATGT R: CTCCTCATTGCCTCCTTTTC	60	FJ_860004
*cat*	F: ATGGTGTGGGACTTCTGGAG R: AGTGGAACTTGCAGTAGAAACG	60	FJ_860003
*gpx*	F: AGTTAATCCGGAATTCGTGAGA R: TGAGTGTAGTCCCTGGTTGTTG	60	FM_013606.1c
*gshr*	F: TGCACCAAAGAACTGCAGAA R: ACGAGTGTCACCTCCAGTCC	60	FM_020412
*ef1a*	F: CAAGGAGGGCAATGCCAGT R: GAGCGAAGGTGACGACCAT	60	XM_051391262
*rpl13a*	F: TCTGGAGGACTGTCAGGGGCATGC R: AGACGCACAATCTTGAGAGCAG	60	XM_051389730
*rpl17*	F: TTGAAGACAACGCAGGAGTCA R: CAGCGCATTCTTTTGCCACT	60	AF_139590.1
*fau*	F: GACACCCAAGGTTGACAAGCAG R: GGCATTGAAGCACTTAGGAGTTG	68	XM_051408030

Ta = annealing temperature in the qPCR. *cox4a* (cytochrome c oxidase), *cs *(citrate synthase), *pgc-1α* (Peroxisome proliferator-activated receptor gamma coactivator 1-alpha), *tfam* (transcription factor A), *ucp2* and *3* (Uncoupling proteins), *fis1 *(fission protein), *mit1 *(mitofusin), *Tmem20* (transmembrane protein 20), *sod *(Superoxide dismutase), *cat *(Catalase), *gpx* (Glutathione peroxidase), *gshr* (Glutathione reductase), *ef1a* (Elongation factor 1-alpha 1), *rpl13* (60S ribosomal protein L13), *rpl17* (Ribosomal protein L17), and *fau* (40S ribosomal protein S30).

**Table 2 animals-14-00274-t002:** Somatic-growth parameters of sea bass voluntarily swimming (Control) or subjected to sustained swimming.

	Control	Exercise
Initial BW (g)	3.93 ± 0.10	3.94 ± 0.11
Final BW (g)	14.76 ± 0.51	13.5 ± 0.46
CF ^a^	1.70 ± 0.04	1.72 ± 0.03
SGR ^b^	3.02 ± 0.09	2.95 ± 0.18
HSI ^c^	1.91 ± 0.10	1.92 ± 0.13
MSI ^d^	38.31 ± 0.69	40.15 ± 0.76
MFI ^e^	3.29 ± 0.23	3.31 ± 0.33

Values are mean ± SEM. *n* = 3 for initial and final body weight (BW) and SGR; *n* = 12 for HSI, MSI and MFI. ^a^ Condition Factor = 100 × BW/TL^3^. ^b^ Specific growth rate = [100 × (ln final BW − ln initial BW)] × days^−1^. ^c^ Hepatosomatic index = g liver × 100 g BW^−1^. ^d^ Muscle–somatic Index = g muscle × 100 g BW^−1^. ^e^ Mesenteric fat index = g fat × 100 g BW^−1^.

**Table 3 animals-14-00274-t003:** Proximal composition and RNA and DNA contents in white muscle of sea bass voluntarily swimming (control) or subjected to sustained swimming.

	Control	Exercise
*Composition*		
Wet weight (%)	76.8 ± 0.15	76.46 ± 1.39
Protein (% w.w)	19.65 ± 0.4	19.57 ± 1.24
Lipids (% w.w)	1.28 ± 0.19	1.55 ± 0.26
Glycogen (% w.w)	0.10 ± 0.02	0.10 ± 0.02
RNA (µg/mg prot.)	14.18 ± 0.98	16.66 ± 0.88
DNA (µg/mg prot.)	7.27 ± 0.68	5.78 ± 0.55
RNA/DNA	2.28 ± 0.22	3.10 ± 0.25 *

Values are mean ± SEM. *n* = 12. Significant differences using Student’s *t*-test: (*) *p* < 0.05.

**Table 4 animals-14-00274-t004:** Isotopic composition (^15^N/^13^C) in white muscle of sea bass voluntarily swimming (control) or subjected to sustained swimming.

	Control	Exercise
δ^13^C-muscle	−19.75 ± 0.12	−19.68 ± 0.16
δ^15^N-muscle	12.32 ± 0.04	12.04 ± 0.06 ***
δ^13^C-protein	−19.81 ± 0.02	−19.84 ± 0.03
δ^15^N-protein	12.59 ± 0.08	12.51 ± 0.03
δ^13^C-glycogen	−23.15 ± 0.33	−23.39 ± 0.20
Δ^15^N-muscle ^1^	1.97 ± 0.04	1.69 ± 0.06 ***

Values are mean ± SEM. *n* = 12. ^1^ Δ^15^N is the nitrogen fractionation. Significant differences using Student’s *t*-test: (***) *p* < 0.001.

**Table 5 animals-14-00274-t005:** Aerobic capacity of sea bass after 6 weeks of voluntary swimming (Control) or being subjected to sustained swimming.

	Control	Exercise
RMR ^1^	223.58 ± 26.50	254.25 ± 20.33
MMR ^1^	691.28 ± 34.40	788.55 ± 21.11 *
AMS	467.69 ± 46.32	534.31 ± 18.25

Values are mean ± SEM. *n* = 12. ^1^ mg O_2_·kg·h^−1^. RMR is MO_2_ resting; MMR is MO_2_ max; AMS is Aerobic scope. Significant differences using Student’s *t*-test: (*) *p* < 0.05.

## Data Availability

The original contributions presented in the study are included in the article/Appendix A. Further inquiries can be directed to the corresponding author.

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
