# Peer review of "Improving the Aerobic Capacity in Fingerlings of European Sea Bass (Dicentrarchus labrax) through Moderate and Sustained Exercise: A Metabolic Approach"

_animals, 2024, doi:10.3390/ani14020274_

Round 1

Reviewer 1 Report

Comments and Suggestions for Authors

General comments:

This is a valuable, multi-faceted and well executed study testing how/whether sustained exercise affects growth performance, body composition, and other metabolic parameters in an economically important fish species.  My only general suggestions are that (1) the Introduction could be made more readable by moving some detailed material to the Discussion section, and (2) some of the writing could be clarified (see specific comments).   

Specific comments:

L 16: When the authors say “both muscles” do they mean red and white muscle?  Please clarify.

L 18, 605: Please clarify what is meant by “robustness”.  Robust in what way? In terms of survival, tolerance of stress, ability to grow and reproduce under a wide variety of environmental conditions, or what?

L 39, 605: Physical fitness or reproductive (evolutionary) fitness?

L 45-47: It would be useful to the general reader if “feed efficiency” was defined upon first use.  Otherwise, this sentence is confusing.  If feed efficiency is defined as growth/ingested or metabolized energy, then exercise should decrease it by using energy that would otherwise be available for growth.  However, if feed efficiency is defined as growth (minus costs of maintenance & activity)/ingested energy, then it is possible that it could be increased by exercise.

L 49-50: Insert “of” between “because” and “the”.

L 51: Insert “such” between “salmonids” and “as”.

L 66: Omit “to”.

L 69-71: Actually, the lower bound of energy expenditure is set by the resting metabolic rate, not the routine metabolic rate, which includes activity costs.

L 75: Omit “in”.

L 88-91: Awkwardly worded sentence.  Please clarify.

L 91: Please define all acronyms (e.g., AMS) at first use.

L 131: Should “typified” be “quantified”?

L 462: Omit “in”.

L 467: Change “of” to “for”.

L 469: Change “the growth” to “their growth”.

L 470-473: Awkward wording.

L 482-483: Omit “has been observed that”

L 510: Change “carrying” to “carried”.

L 525: Change “compensates” to “compensate”.

L 536-539: Awkward wording.

L 601-602: Insert “at” between “fingerlings” and “about”. 

L 603: Change “without” to “no”?

Comments on the Quality of English Language

Should be improved (suggestions provided above).

Author Response

Please, you will find attached file.

Reviewer 2 Report

Comments and Suggestions for Authors

The study of Perelló-Amorós et al. discusses the effect of moderate and sustained swimming on optimum aerobic conditions of European seabass. The experiment well designed, and manuscript well prepared. There are few comments could be considered before publication.

The initial weight of the experimental fish in line 141 is different than in line 168.

L 142: incomplete sentence “and in the facilities of the Faculty of Biology”.

L 144 replace “physical” filter by “mechanical” filter

L 150: could you add the details of the used commercial diet.

L 248 correct the reference citation.

L 167: add no for the subtitle and correct the order of other subtitles.

In the footnotes of Table 1. Could you add the full names of different genes.

Could you correct the citation of SPSS 22 as follows: (IBM SPSS Statistics for Windows, Version 22.0. Armonk, NY: IBM Corp.)

Could you change the location of Figure 1 after subtitle 3.1.

Footnote of table 2 and 3 is not correct.

Could you avoid abbreviation in the conclusion.

Comments on the Quality of English Language

fine

Author Response

Please, you will find an attached file
